# Genomic and Transcriptomic Analysis of Hypercholesterolemic Rabbits: Progress and Perspectives

**DOI:** 10.3390/ijms19113512

**Published:** 2018-11-08

**Authors:** Jianglin Fan, Yajie Chen, Haizhao Yan, Baoning Liu, Yanli Wang, Jifeng Zhang, Y. Eugene Chen, Enqi Liu, Jingyan Liang

**Affiliations:** 1Department of Molecular Pathology, Faculty of Medicine, Graduate School of Medical Sciences, University of Yamanashi, Yamanashi 409-3898, Japan; lywchen@163.com (Y.C.); yanhaizhao@126.com (H.Y.); 2Department of Pathology, Xi’an Medical University, Xi’an 710021, China; yanliwang@xiyi.edu.cn; 3Research Institute of Atherosclerotic Disease and Laboratory Animal Center, School of Medicine, Xi’an Jiaotong University, Xi’an 710061, China; liubaoning88@stu.xjtu.edu.cn (B.L.); liuenqi@mail.xjtu.edu.cn (E.L.); 4Center for Advanced Models for Translational Sciences and Therapeutics, University of Michigan Medical Center, Ann Arbor, MI 48109, USA; jifengz@med.umich.edu (J.Z.); echenum@med.umich.edu (Y.E.C.); 5Institute of Translational Medicine, Medical College, Yangzhou University, Yangzhou 225001, China; 6Jiangsu Key Laboratory of Integrated Traditional Chinese and Western Medicine for Prevention and Treatment of Senile Diseases, Yangzhou University, Yangzhou 225001, China; 7Jiangsu Co-Innovation Center for Prevention and Control of Important Animal Infectious Disease and Zoonoses, Yangzhou 225001, China

**Keywords:** hypercholesterolemia, fatty liver, rabbit, transcriptome

## Abstract

Rabbits (Oryctolagus cuniculus) are one of the most widely used animal models for the study of human lipid metabolism and atherosclerosis because they are more sensitive to a cholesterol diet than other experimental animals such as rodents. Currently, two hypercholesterolemic rabbit models are frequently used for atherosclerosis studies. One is a cholesterol-fed wild-type rabbit and the other is the Watanabe heritable hyperlipidemic (WHHL) rabbit, which is genetically deficient in low density lipoprotein (LDL) receptor function. Wild-type rabbits can be easily induced to develop severe hypercholesterolemia with a cholesterol-rich diet due to the marked increase in hepatically and intestinally derived remnant lipoproteins, called β-very low density lipoproteins (VLDL), which are rich in cholesteryl esters. WHHL rabbits are characterized by elevated plasma LDL levels on a standard chow diet, which resembles human familial hypercholesterolemia. Therefore, both rabbit models develop aortic and coronary atherosclerosis, but the elevated plasma cholesterol levels are caused by completely different mechanisms. In addition, cholesterol-fed rabbits but not WHHL rabbits exhibit different degrees of hepatosteatosis. Recently, we along with others have shown that there are many differentially expressed genes in the atherosclerotic lesions and livers of cholesterol-fed rabbits that are either significantly up- or down-regulated, compared with those in normal rabbits, including genes involved in the regulation of inflammation and lipid metabolism. Therefore, dietary cholesterol plays an important role not only in hypercholesterolemia and atherosclerosis but also in hepatosteatosis. In this review, we make an overview of the recent progress in genomic and transcriptomic analyses of hypercholesterolemic rabbits. These transcriptomic profiling data should provide novel insight into the relationship between hypercholesterolemia and atherosclerosis or hepatic dysfunction caused by dietary cholesterol.

## 1. Introduction

Hypercholesterolemia, atherosclerosis and hepatosteatosis, such as non-alcohol steatosis hepatitis and non-alcohol fatty liver disease, are life-style-related metabolic disorders, and are major health problems in modern society [1,2]. Although each disease can be caused by different mechanisms, there are many similarities and they are often closely related with each other in certain aspects. For example, hypercholesterolemia is the major risk factor for atherosclerosis, but it is also associated with hepatosteatosis [3]. On the other hand, the liver limits cholesterol synthesis, and hepatic dysfunction can cause hyperlipidemia and may affect atherosclerosis via increased production of several cytokines in the circulation [4]. To elucidate the molecular mechanisms and develop therapeutic strategies for these diseases, appropriate experimental animals are often used. Currently, mice, especially genetically modified mice, are the most common animal model. Other models such as rats, rabbits, pigs and nonhuman primates are also applied. In particular, rabbit models are widely used due to their high sensitivity to dietary cholesterol and rapid development of atherosclerosis [5]. Cholesterol-fed rabbits are also used for the study of glucose metabolism [6], insulin resistance [7,8] and fatty liver [9]. Compared with rodents (rats and mice), rabbits have several unique metabolic features that are similar to humans. For example, rabbits have abundant cholesteryl ester transfer protein (CETP) activity in the plasma as do humans [10], whereas mice and rats are deficient in plasma CETP. Furthermore, as rabbits do not have hepatic apolipoprotein-(apo)B mRNA editing activity, rabbit apoB-48 is only present in intestinally derived chylomicrons, similar to humans, but unlike mice and rats in which apoB-48 is contained in both chylomicrons and hepatically derived VLDL and LDL. Therefore, rabbits provide a unique system to study the effects of dietary cholesterol on hypercholesterolemia, atherosclerosis and liver dysfunction. However, genomic information for rabbits has long been lacking and transcriptomic information for rabbits related to atherosclerosis and liver dysfunction has not been fully elucidated, which hampers the use of rabbits for gene-targeting and translational research. The emergence of next-generation sequencing platforms has made genomic and transcriptomic analyses affordable for many researchers. In this review, we provide an overview of recently published rabbit genome information and transcriptomic profiling of atherosclerotic lesions and livers and discuss how to use such information for future studies of metabolic diseases.

## 2. Rabbit Genome Information

The Human Genome Project was completed by an international team on 14 April 2003 (https://www.genome.gov/11006943/human-genome-project-completion-frequently-asked-questions/). After that, mouse [11] and rat [12] genomes were sequenced in succession, which greatly advanced scientific research using genetically modified mice, such as knock-out (KO) mice, to study human diseases. On the other hand, rabbit genomic information has long been lacking mainly due to budget shortages and limited research communities. In 2004, the National Human Genome Research Institute initially approved the rabbit genome for 2× “light” genome coverage sequencing at the Broad Institute. As this genomic information is still limited, Drs. Yuka Manabe (Johns Hopkins University School of Medicine), Rose Mage (NIH) and Jean Chang (Broad Institute of Massachusetts Institute of Technology and Harvard University) made great efforts to submit a “white paper” to push for more extensive coverage of the rabbit genome (personal communication). After an international roundtable discussion held at the Broad Institute in 2006, the Rabbit Genome Project was initiated. In 2014, Carneiro and associates first reported a high-quality reference genome for the European rabbit with references to domestication and speciation [13,14]. At the same time, we organized another International Rabbit Genome Sequencing Project Consortium consisting of researchers from the US, Japan and China aiming at launching more extensive whole-genome sequencing of three laboratory rabbit lines [Japanese white (JW) rabbits, New Zealand white (NZW) rabbits and Watanabe heritable hyperlipidemic (WHHL rabbits)], in addition to deep transcriptome sequencing of the aortas, livers, hearts and kidneys of cholesterol-fed NZW and WHHL rabbits [15]. After a two-year effort, we successfully completed whole-genome sequencing of 10 male rabbits for each of the three lines, resulting in a depth of coverage of approximately 13× for each individual after alignment to the reference genome. In total, we identified 29.8 million single nucleotide polymorphisms (SNPs) and 1.6 million small indels in the 30 genomes. In this study, we were particularly interested in WHHL rabbits because they were bred in a closed colony since they were established in 1980 [16] and exhibited a number of metabolic abnormalities such as insulin resistance [17] and visceral fat accumulation [18] in addition to spontaneous hypercholesterolemia and atherosclerosis [19]. We envisioned other deleterious mutations to arise at a high frequency by genetic drift due to the extremely small population size of WHHL rabbits. To search for such deleterious mutations possibly involved in their phenotypes, we compiled a comprehensive gene list associated with cardiovascular diseases from both the knowledge database and human genome-wide association studies [15]. We identified 24 putative deleterious mutations enriched in WHHL rabbits. Among these mutations, aldehyde dehydrogenase-2 (ALDH2) mutation (substitution-missense, position 99, R→C) (R99C) in WHHL rabbits looks like a candidate mutation because its allele frequency was low or none in the normal wild-type rabbits (10% in NZW and 0% in JW vs. 100% in WHHL). ALDH2 is a mitochondrial enzyme detoxifying acetaldehyde and endogenous lipid aldehydes and previous studies showed that a loss-of-function mutation of ALDH2 in humans causes alcohol flushing and is associated with high risk of cardiovascular diseases [20,21], suggesting that ALDH2 plays a protective role in cardiovascular diseases. A recent study using ALDH2 KO mice revealed that ALDH2 regulates foam cell formation through interactions with LDL receptors and 5’ adenosine monophosphate-activated protein kinase (AMPK) pathway [22], providing a new molecular mechanism for ALDH2 in atherosclerosis. Therefore, these results suggest that the deleterious mutations possibly function as genetic modifiers involved in the pathophysiology of WHHL rabbits and also genomic analysis may provide another novel approach to identify putative gene mutations associated with different diseases. However, whether these genetic modifiers are directly involved in the phenotypes of WHHL rabbits is still unknown. Regardless of this, with the accomplishment of rabbit genome sequencing [13,14,15], it has become much easier to design PCR primers to study gene expression in rabbits and to generate KO rabbits using novel gene-editing technologies [zinc-finger nucleases (ZFN), transcription activator-like effector nucleases (TALENs), clustered regularly interspaced short palindromic repeats CRISPR-associated proteins 9 (CRISPR-Cas9)] in recent years [23]. Rabbit genome information is now available from the National Center for Biotechnology Information (NCBI) database, and a comprehensive database containing both rabbit genome and transcriptome information has been comprehensively constructed by the Chinese Academy of Sciences [24], which can be reached at http://www.picb.ac.cn/RabGTD/.

## 3. Transcriptome Profiling of Rabbit Atherosclerotic Lesions 

Availability of rabbit genome information makes it possible to perform a transcriptomic analysis of atherosclerosis in rabbits. We conducted deep transcriptome sequencing of aortas and livers collected from both wild-type male (JW and NZW) rabbits and hypercholesterolemic (cholesterol-fed and WHHL) rabbits, and then performed RNA expression analyses for cholesterol-fed NZW versus chow-fed NZW rabbits, chow-fed WHHL versus chow-fed JW rabbits and cholesterol-fed NZW rabbits versus chow-fed WHHL rabbits. In atherosclerosis research using rabbits, aortas are mostly used for pathological observation and quantitation of atherosclerosis because aortas are susceptible to atherosclerosis and easy to quantify gross and microscopic lesions [5]. Representative micrographs of aortic atherosclerosis were shown in Figure 1A. Transcriptomic analysis revealed that in aortas of cholesterol-fed rabbits and WHHL rabbits, there were 2719 and 1627 differentially expressed genes (DEGs) compared with those of normal wild-type rabbits [15]. However, based on a threshold (greater than a 2-fold increase or decrease versus control with *p* < 0.05), it is found that the transcriptional response patterns in the aortas of cholesterol-fed and WHHL rabbits are essentially similar (Figure 1B). Among these DEGs, genes associated with inflammatory responses were highly upregulated such as cytokines and chemokines along with their receptors and matrix metalloproteinases (MMPs) (Figure 1C) which is similar to those of gene expression changes in the lesions of atherosclerosis in mice [25,26,27]. In human studies, transcriptomic profiling revealed that proprotein convertase subtilisin/kexin type 6 is high expressed in the unstable plaques [28]. These results further strengthened the notion that atherosclerosis is essentially evoked by inflammatory reaction [2]. Although transcriptomic profiling analysis usually digs out numerous DEGs, it is almost impossible to elucidate pathophysiological roles of each DEG in the process of disease. Therefore, we need to focus on some specific target genes to find a clue for the future research. One of such efforts is to analyze those genes which are possibly involved in the plaque vulnerability. In human atherosclerotic lesions, up-regulated matrix metalloproteinases (MMP) and/or down-regulated tissue inhibitors of metalloproteinase (TIMP) have be implicated in the plaque rupture or erosion [29]. Toward this end, we re-analyzed the transcriptomic profiling database [15] with special reference to MMPs and TIMPs in the aortas of hypercholesterolemic rabbits because MMPs are involved in lesion formation and possibly in plaque rupture in rabbits [30]. Eighteen types of MMPs and three TIMPs were detected in the aortas of rabbits, but four MMPs were markedly and uniquely upregulated in the lesions of hypercholesterolemic rabbits (Figure 2). The most notable change among these four MMPs was in MMP-12, an elastase, whose expression was upregulated by up to 6000-fold in WHHL and 18,000-fold in cholesterol-fed rabbits compared with in chow-fed control rabbits, suggesting that MMP-12 plays an important role in the pathogenesis of atherosclerosis. This finding is also consistent with our previous study which showed that overexpression of MMP-12 in macrophages significantly increased aortic lesions in transgenic rabbits [31]. In the arterial wall, three MMPs with interstitial collagenase activity have been demonstrated to play an important role in atherosclerosis [32,33]. As shown in Figure 2, the expression of MMP-13, an interstitial collagenase, in the aorta was increased by 300-fold in WHHL rabbits and 160-fold in cholesterol-fed rabbits compared with in control rabbits. It has been reported that selective inhibition of MMP-13 increases the collagen content in the lesions in mice [34]. Thus, future investigation into whether MMP13 also plays a critical role in atherosclerosis and plaque rupture in rabbits will be of interest. Interestingly, MMP-8 expression was not detected in rabbit aortic lesions, but this is expected because MMP-8 is secreted by neutrophils which is not present in rabbit lesions, suggesting that MMP-8 is not involved in the pathogenesis of rabbit atherosclerosis [33]. MMP-1, another important collagenase, was not expressed in the aortas of chow-fed control rabbits, and only expressed in the aortic lesions of both WHHL and cholesterol-fed rabbits. Of note, mice do not possess a homologue for MMP-1; therefore, MMP-1 does not affect mouse atherosclerosis [35]. We recently generated transgenic rabbits expressing human MMP-1 in a macrophage lineage and found that increased MMP-1 expression played a functional role in aortic aneurysm formation (Niimi and Fan, unpublished data). MMP-3, stromelysin, was not expressed in the normal aortas, but was expressed in aortic lesions, similar to MMP-1. MMP-3 can activate MMP-9, gelatinase B [36], which was also upregulated by 55-fold in WHHL rabbits and 128-fold in cholesterol-fed rabbits compared with that in control rabbits. Increased expression of MMP-9 was detected in human atherosclerosis [37,38], but the functional roles of MMP-9 in atherosclerosis remain unclear from mouse studies. In one report, MMP-9 deficiency protected against cholesterol diet-induced atherosclerosis [39], but in another, MMP-9 inactivation increased the atherosclerotic plaque growth and progression [40]. We generated transgenic rabbits expressing MMP-9 in a macrophage lineage and found that MMP-9 overexpression may enhance vascular calcification (Chen and Fan, unpublished data). Therefore, most MMP functions in atherosclerosis are still unclear and MMPs may have functions other than merely hydrolyzing the extracellular matrix. In addition to MMPs, an endogenous tissue inhibitor of metalloproteinases, TIMP-1, was also upregulated by 4-fold in WHHL rabbits and 5-fold in cholesterol-fed rabbits compared with in control rabbits, consistent with the previous study [41]. Increased expression of TIMP-1 reduced atherosclerosis in apolipoprotein (apo)-E KO mice [42], whereas TIMP-1 deficiency increased medial degradation but did not affect atherosclerosis in apo-E KO mice [43]. The overall increase (fold) in the expression of TIMP-1 was relatively lower than that of the four MMPs in the aorta, suggesting that TIMP-1 is mainly expressed by vascular smooth muscle cells instead of infiltrating macrophages. However, DEG patterns in the aortic lesions were similar between WHHL and cholesterol-fed rabbits as mentioned above, suggesting that regardless of the different types of hypercholesterolemia in WHHL (predominated by LDLs) and cholesterol-fed (predominated by β-VLDLs or remnant lipoproteins) rabbits, the process of lesion development occurs via a similar pathway, namely, inflammation initiated by lipid deposition in the intima [44,45].

## 4. Transcriptomic Profiling of Livers 

As the liver is an important organ for mediating lipid metabolism, we further investigated transcriptional changes in the livers of these two hypercholesterolemic rabbits by RNA sequencing analysis. As WHHL rabbits exhibit no histological changes in the liver, there were no significant changes in the DEGs compared with control rabbits. On the contrary, cholesterol-fed rabbits often have varying degrees of fatty liver changes [9] (Figure 3). RNA sequencing analysis revealed 14,413 DEGs in the livers of cholesterol-fed rabbits compared with control NZW rabbits [15]. To make a rigorous analysis of DEGs, we re-analyzed the transcriptomic profiling database of the livers [15]. There are totally 967 DEGs with 590 upregulated and 377 downregulated in cholesterol-fed rabbits but 132 DEGs were clearly associated with hepatosteatosis (Figure 4). These DEGs can be classified into five categories according to their functions, including inflammation, cell proliferation, apoptosis, lipid metabolism and cell channels (Figure 4). These DEGs suggest an inflammatory response and abnormal lipid metabolism are essentially present in the liver of rabbits fed a cholesterol diet [9,46]. As fatty liver is not observed in WHHL rabbits, it suggests that it is exogenous dietary cholesterol that elevates the plasma levels of cholesterol and induces hepatic dysfunctions. Fatty liver in rabbits can be induced by feeding a “high” (1% w/w) cholesterol diet for two weeks [46] or “low” (0.3% w/w) cholesterol diet for 16 weeks [9], and these rabbit livers exhibit typical pathological features of hepatosteatosis. This phenomenon was also reported in other animals such as rats and mice fed a high-fat and cholesterol diet. In high-fat fed Wistar rats, the hepatic fatty acid utilization through β-oxidation was inhibited whereas lipogenesis was enhanced [47]. In apoE*3 Leiden mice, high-fat diet altered hepatic lipid metabolism and inflammatory response [48] and similar findings were also seen in C57/BL6 mice [49]. In a separate study, we also found that sterol regulatory element binding protein 1 was upregulated by up to 40-fold in the liver of cholesterol-fed rabbits [50], and endoplasmic reticulum stress may also be involved in fatty liver development [9]. Although RNAseq studies have generated a marked amount of DEGs and predicted many molecular pathways, their pathophysiological significance needs to be confirmed in the future using appropriate experimental strategies. Moreover, there are many differences in these DEGs between animals and humans [51].

## 5. Concluding Remarks

Genomic and transcriptomic sequencing analyses have provided a novel platform to disclose a number of molecular pathways that may be responsible for or associated with the metabolic diseases such as atherosclerosis [15], hepatosteatosis [50] shown in hypercholesterolemic rabbits. Elucidation of MMPs and TIMPs in the lesions of atherosclerosis may help understand the molecular mechanisms of plaque rupture. Interestingly, hypercholesterolemic rabbits exhibit certain degrees of cognitive deficits and a recent study performed a transcriptomic analysis of the brains of these rabbits [52]. However, rabbits are out-bred and often exhibit different responses to varying stimuli. For example, on the same cholesterol diet, there is a gender difference in the same wild-type rabbits [53] and some rabbits even do not exhibit marked hypercholesterolemia, termed “low responders” [54]. This issue is frustrating because the presence of low responder rabbits in cholesterol-fed rabbit experiments generates large variations in data. The genetic mechanisms for these low responder rabbits are still unknown although enhanced 7α-hydroxylase activity has been proposed [55]. In a future study, the genetic basis of these low responders awaits further investigation. Transcriptomic profiling of the aortic atherosclerosis and livers has provided new insights into the pathogenesis of atherosclerosis and hepatosteatosis; however, other analyses are required. One such analysis is that of long non-coding RNAs (lncRNAs). lncRNAs have been emerging as important regulators in many pathophysiological processes, but the roles of lncRNAs in atherosclerosis are still not well understood [56]. As such, studies to clarify the function of lncRNAs in the lesions in cholesterol-fed rabbits should be performed. In addition to the lesions of atherosclerosis and fatty livers, detecting biomarkers in plasma of hypercholesterolemic rabbits have been attempted using mass spectrometry. Johno et al. recently successfully established a method using probe electrospray ionization mass spectrometry to detect plasma metabolites and found that in the plasma of hypercholesterolemic rabbits, the major metabolites are cholesterol sulfate and phosphatidylethanolamine(PE)[PE18:0/20:4], which may become promising new biomarkers of atherosclerosis [57]. Bai et al. conducted a whole-plasma *N-*glycan profiling of hypercholesterolemic rabbits along with hypercholesterolemic patients by electrospray ionization mass spectrometry and showed that there are increased plasma levels of high-mannose and high complex/hybrid *N*-glycan levels [58]. Therefore, it can be predicted that in the future, with emergence of these new technologies, we will face enormous “-omic” or “profiling” data but eventually we need to verify their pathophysiological significance in human diseases.

## Figures and Tables

**Figure 1 ijms-19-03512-f001:**
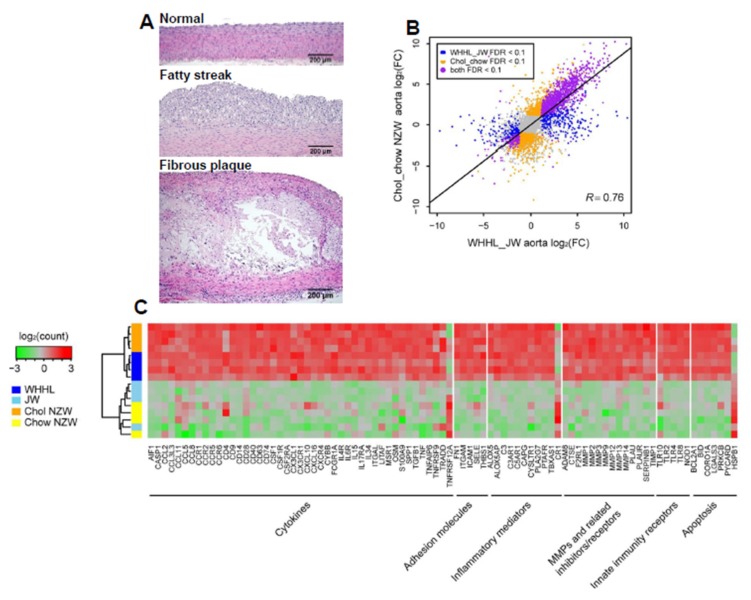
Representative micrographs of rabbit aortic atherosclerosis (**A**). Normal (**top**), fatty streak (**middle**) and fibrous plaque (**bottom**). Aortic lesions of atherosclerosis were stained with hematoxylin and eosin staining. Transcriptome profiling of rabbit models with aortic atherosclerosis (**B**). Transcriptomic profiling of aortas from chow-fed NZW and cholesterol-fed NZW, and chow-fed JW and Watanabe heritable hyperlipidemic (WHHL) rabbits were conducted and a strong positive correlation of expression changes in the aorta between cholesterol-fed and WHHL rabbits is shown in (**B**). FDR: false discovery rate, FC: fold change. The correlation coefficient was calculated for differentially expressed genes (DEGs) in at least one condition. Heatmap of representative DEGs responsible for inflammation responses in the aorta was shown (**C**). The read counts were log-transformed and normalized across samples (**C**). (**B**,**C**) are modified from the original published figures [15].

**Figure 2 ijms-19-03512-f002:**
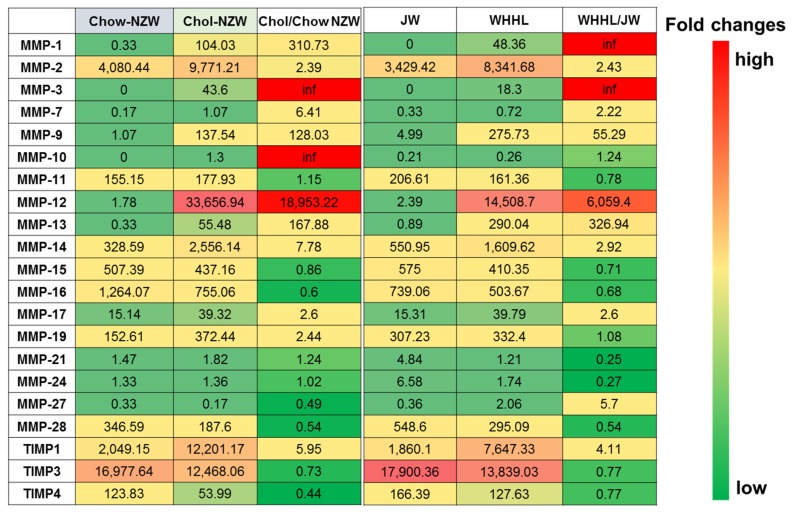
Heatmap of matrix metalloproteinases (MMPs) and tissue inhibitors of metalloproteinase (TIMPs) in the lesions of aortas of NZW rabbits (chow-fed versus cholesterol-fed) (left) and JW rabbits (JW wild-type versus WHHL rabbits). Average reads of four animals from each group or ratios of NZW cholesterol-fed/NZW chow-fed or WHHL/JW-chow fed are shown. Marked increases in expression were noted for MMP-12, MMP-13, MMP-9 and TIMP-1. RNA expression of MMP-1 and MMP-3 were not detected in the normal aorta. Inf means infinitive.

**Figure 3 ijms-19-03512-f003:**
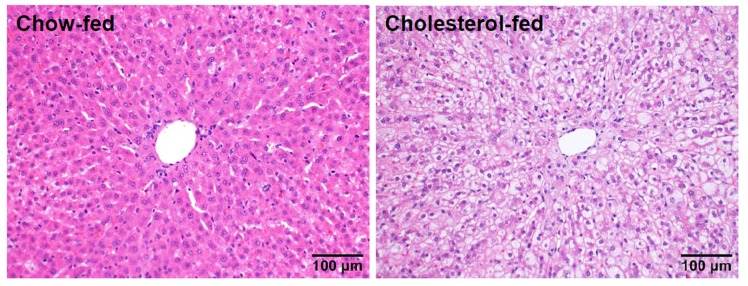
Histological images of livers of normal (**left**) and cholesterol-fed rabbits (**right**). Paraffin sections were stained by hematoxylin and eosin staining, and livers of cholesterol-fed rabbits exhibit features of steatosis with increased lipids in the cytoplasm.

**Figure 4 ijms-19-03512-f004:**
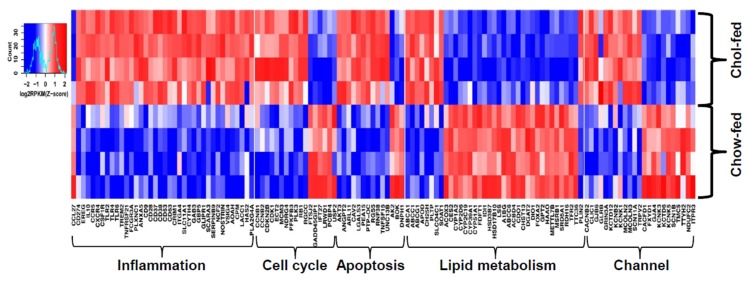
Heatmap of DEGs in the livers of chow-fed and cholesterol-fed rabbits. Four male rabbits from each group were analyzed and individual data were shown. These DEGS were classified into four groups: 39 DEGs are related with Inflammation including receptors, ligands, membrane proteins, cytoplasm proteins and enzymes; 33 DEGs related with cell growth and apoptosis, 36 DEGs related with lipid metabolism; 24 DEGs related with cell membrane channels. The color brightness in heatmap indicate the normalized z-scores of log2-transformed gene expression values. Red color represents for a relatively high expression value whereas blue color for low expression value. n = 4 for each group. EREG; epiregulin, NOSTRIN; nitric oxide synthase trafficking, AOAH; acyloxyacyl hydrolase, CPM; carboxypeptidase M, RGCC; regulator of cell cycle, PTP-OC; protein tyrosine phosphatase, receptor type O, BOK; bcl-2 related ovarian killer, APOD; apolipoprotein D, PLTP; phospholipid transfer protein, LSS; lanosterol synthase, HAAO; 3-hydroxyanthranilate 3,4-dioxygenase, GLRB; glycine receptor beta.

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
