# Peer review of "Genomic and Transcriptomic Analysis of Hypercholesterolemic Rabbits: Progress and Perspectives"

_ijms, 2018, doi:10.3390/ijms19113512_

Round 1
Reviewer 1 Report
This is a review of some recent experiments conducted to elucidate genomic and transcriptomic profiling of rabbits. The original studies were intended to model diet-induced and familial hypercholersterolemia and related diseases (eg hepatosteatosis) in humans. The review initially does a fine job of explaining the utility of rabbit models in this area and the need for more genomic and transcriptomic profiling in rabbits.
The abstract makes it seem as if the authors are reporting original research, but the first section clearly states this is a review (line 67). I assume the studies presented here have been published in other journals but citations are lacking in several places. If these results have not been previously published, this must be clearly stated. Without this information, it is difficult to evaluate the methodology behind the presented information. It is also unclear if the authors were involved in the reported research, or if this is simply meant to be a literature review.
Reply: Thanks for your comments. In the abstract, we have stated that this is a review article in lines 24-26. In addition, we have double-checked all references and added new references (16-18, 25-29, 47, 49, 51-53, 57). In the revised manuscript, we have clearly mentioned the work performed by ourselves and work reported by others. To make it clear that this is a review article, we have modified the title.
In section 3, experiments are described which evaluate both diet (high cholesterol chow or normal chow) and rabbit strain (WHHL, NZ, JW). The authors state "We found that in aortas of cholesterol-fed rabbits and WHHL rabbits, there were 2,719 and 1,627 differentially expressed genes (DEGs) compared with those of normal rabbits." The exact meaning of this sentence is unclear. Please specify exactly what is meant by 'normal rabbits'. I assume this refers to NZ and JW rabbits, but it is unclear if this refers to rabbits fed HC chow, normal chow, or both. I suggest using paragraphs to break up the text in this section.
Reply: Thanks for your critiques. We have rephrased the sentence in the revised manuscript. “Normal rabbits” means chow-fed JW and NZW rabbits.
In section 4: Columns are unlabeled in Figure 4B, 4D. It is unclear what the columns represent in the Figure 4 heatmaps. Individual animals? The legend does not provide enough information for the reader to understand what information is being presented.
Reply: We apologized for this mistake. Columns are labeled and explained in the legend of the revised manuscript. Each data represents individual data.
Section 5 needs a summary of how the work presented in the review contributes to the current status of the field. As it stands, section 5 only presents directions for future research.
Reply: Thanks for your suggestion and we have modified section 5 to indicate how the current work contributes to the future in the field.
Reviewer 2 Report
This review by Fan et al. describes the importance of the rabbit animal model for lipid metabolism and atherosclerosis research. The manuscript in particular focuses on a recent study by this group (Wang et al. Sci Reports 2016) describing genomic and transcriptomic sequencing of 3 rabbit models (JW, NZW and WHHL) with hypercholesterolemic or regular chow diets. While this is an important effort to champion this model for these kinds of studies, this review is more of a reanalysis of this initial research effort, with some overlap of findings (some of the wording is verbatim from the original article, in particular the section on the rabbit genome information, hence my plagiarism flag). Therefore it is not really clear from the current version of this paper if this should be classified as a review given the major focus on this single article from their group or an analysis effort to glean additional details not outlined in the 2016 article.
Major issues:
1) As mentioned, the "Rabbit genome information" section is essentially a rehash of the previous article. While there is some background on previous efforts to sequence the rabbit genome, the "new" information regarding putative deleterious SNPs and other mutations are the same that were already published, without any additional insight. This sections needs to be either expanded with new information with review of other current efforts or only mentioned briefly, with major focus on the transcriptomic information.
Reply: Thanks for your critical suggestion and we have revised this part and emphasized what we learnt from re-analysis of the lesions and livers.
2) The transcriptomic analysis (which appears to be a re-analysis or an alternative focus) has a little bit more insight on MMP and TIMP regulation in the aorta dataset, and a more in depth functional analysis of DEGS in the steatotic liver of cholesterol-fed NZW rabbits. However much of the focus is still in general on the inflammatory response in these models, which is what was mentioned in the 2016 report. Therefore the new information in this manuscript is fairly minimal.
It is suggested that a full re-analysis be performed with different methods and interpretation packages, and methods fully outlined, to provide an original, impactful piece. As is, this is hybrid review/analysis article with minimal novelty as either.
Reply: We greatly appreciate your critiques in this regard. We have revised these parts and emphasized what we have learnt from re-analysis. For the lesions of atherosclerosis, we focused on MMPs and TIMPs because it is very important for use to understand their pathophysiological significance in the plaque rupture. In addition, we have discussed other information in this field including metabolic and N-glycan profiling of cholesterol-fed rabbit plasma.
Other minor concerns/suggestions:
1) Were male or female rabbits used? Gender may have interesting impacts on response to lipid metabolism perturbation, potentially mediated through Stat5b and association of Ppar-gamma (also seen in these data) in mice (see PMID: 26959237). It may be an interesting angle to tackle in the rabbit model if this data is available. If kept as a review, this could be an interesting discussion.
Reply: All animals used in these studies were male (Page 4, Line 31) because it is well known that there is a gender difference in rabbits in terms of plasma cholesterol levels and their response to a cholesterol diet in the previous publication (Ref.51). This has been explained by influence of sex hormones. For this reason, male rabbits are often used in atherosclerosis studies. We have mentioned this point in the text regarding gender difference (Page 4, Line 31 and Page 17 Line 33).
2)Was there a reason that the JW rabbits were used as a "baseline" comparator for the WHHL rabbits instead of NZW? It would seem to be a more straightforward approach to use the NZW chow-fed so that it could have the same foundation as the cholesterol-fed NZW rabbits, but there may be lineage information that I am missing that is not clear in the manuscript.
Reply: There is a misunderstanding in the literature that WHHL rabbits were originated from NZW rabbits. In fact, WHHL rabbits were from JW rabbits established by Watanabe, Y. in 1980 (Ref.16). That is why we used JW rabbits as a control.
3) On pg.4 line 169. Why would a lower aortic TIMP-1 expression increase compared to 4 MMPs indicate vascular smooth muscle cells instead of macrophages? Need more information that given.
Reply: We agree with you that increased TIMP-1 expression may be from smooth muscle cells rather than macrophages. We have added this point in the text (Page 8, Lines 27-30).
4) Was 2-fold change cut-off applied to both liver and aortic transcriptomic analyses? This pipeline should be consistent or clear why it was different.
Reply: The same standard (2-fold change cut-off with p<0.05) was used in both livers and aortic lesions. Data shown in Figure 4 (livers) was generated using a different software (aortic lesions).
5) A,B, and C labels missing for Fig 1.
Reply: We double-checked labels in Fig.1.
6) Labeling on Fig 4 needs to be fixed. Unclear which samples are control and cholesterol-fed. Also labels need to be given for A, B, C and D.
We apologized for this mistake and Fig.4 was revised along with the legends.
Round 2
Reviewer 1 Report
My concerns have been addressed.
Author Response
Thank you so much.
Reviewer 2 Report
Overall, much improved since last version. I am still having issues with Lines 94-111 as a lot of the text has similar flow and words from the "Deleterious mutations in WHHL rabbits" section (and little bit before) on the 2016 Sci Reports article. I suggest more succinctly summarizing these lines to avoid this replication and setting up the new analysis in the transcriptomic section.
Otherwise this is a nice explanation of the importance of the rabbit model for lipid metabolism and atherosclerosis research and novel gleanings from the transcriptomic data previously generated.
Author Response
Overall, much improved since last version. I am still having issues with Lines 94-111 as a lot of the text has similar flow and words from the "Deleterious mutations in WHHL rabbits" section (and little bit before) on the 2016 Sci Reports article. I suggest more succinctly summarizing these lines to avoid this replication and setting up the new analysis in the transcriptomic section. Otherwise this is a nice explanation of the importance of the rabbit model for lipid metabolism and atherosclerosis research and novel gleanings from the transcriptomic data previously generated. Reply: Thanks for your comments. We have revised this part (Lines 98-110) as below and added new references (22-23). “We identified 24 putative deleterious mutations enriched in WHHL rabbits. Among these mutations, aldehyde dehydrogenase-2 (ALDH2) mutation R99C in WHHL rabbits looks like a candidate mutation because its allele frequency was low or none in the normal wild-type rabbits (10% in NZW and 0% in JW vs. 100% in WHHL). ALDH2 is a mitochondrial enzyme detoxifying acetaldehyde and endogenous lipid aldehydes and previous studies showed that a loss-of-function mutation of ALDH2 in humans causes alcohol flushing and is associated with high risk of cardiovascular diseases(20, 21), suggesting that ALDH2 plays a protective role in cardiovascular diseases. A recent study using KO mice revealed that ALDH2 regulates foam cell formation in atherosclerosis through interactions with LDL receptors and AMP-activated protein kinase(22), providing a new molecular mechanism for ALDH2 in atherosclerosis. Therefore, these results suggest that the deleterious mutations possibly function as genetic modifiers involved in the pathophysiology of WHHL rabbits and also genomic analysis may provide another novel approach to identify putative gene mutations associated with different diseases.”
